# Genome-Wide Identification of Maize Protein Arginine Methyltransferase Genes and Functional Analysis of *ZmPRMT1* Reveal Essential Roles in *Arabidopsis* Flowering Regulation and Abiotic Stress Tolerance

**DOI:** 10.3390/ijms232112793

**Published:** 2022-10-24

**Authors:** Qiqi Ling, Jiayao Liao, Xiang Liu, Yue Zhou, Yexiong Qian

**Affiliations:** Anhui Provincial Key Laboratory of Conservation and Exploitation of Important Biological Resources, College of Life Sciences, Anhui Normal University, Wuhu 241000, China

**Keywords:** histone methylation, protein arginine methyltransferase, *Zea mays* L., abiotic stress, functional analysis

## Abstract

Histone methylation, as one of the important epigenetic regulatory mechanisms, plays a significant role in growth and developmental processes and stress responses of plants, via altering the methylation status or ratio of arginine and lysine residues of histone tails, which can affect the regulation of gene expression. Protein arginine methyltransferases (PRMTs) have been revealed to be responsible for histone methylation of specific arginine residues in plants, which is important for maintaining pleiotropic development and adaptation to abiotic stresses in plants. Here, for the first time, a total of eight *PRMT* genes in maize have been identified and characterized in this study, named as *ZmPRMT1-8*. According to comparative analyses of phylogenetic relationship and structural characteristics among *PRMT* gene family members from several representative species, all maize 8 PRMT proteins were categorized into three distinct subfamilies. Further, schematic structure and chromosome location analyses displayed evolutionarily conserved structure features and an unevenly distribution on maize chromosomes of *ZmPRMT* genes, respectively. The expression patterns of *ZmPRMT* genes in different tissues and under various abiotic stresses (heat, drought, and salt) were determined. The expression patterns of *ZmPRMT* genes indicated that they play a role in regulating growth and development and responses to abiotic stress. Eventually, to verify the biological roles of *ZmPRMT* genes, the transgenic *Arabidopsis* plants overexpressing *ZmPRMT1* gene was constructed as a typical representative. The results demonstrated that overexpression of *ZmPRMT1* can promote earlier flowering time and confer enhanced heat tolerance in transgenic *Arabidopsis*. Taken together, our results are the first to report the roles of *ZmPRMT1* gene in regulating flowering time and resisting heat stress response in plants and will provide a vital theoretical basis for further unraveling the functional roles and epigenetic regulatory mechanism of *ZmPRMT* genes in maize growth, development and responses to abiotic stresses.

## 1. Introduction

In eukaryotic, nucleosome is largely comprised of 146–147 base pairs of DNA and a histone octamer, including four types of histones (namely H2A, H2B, H3 and H4) [1]. The N-terminal tails of these histones can be subjected to post-translational and covalent modifications, including methylation, acetylation, phosphorylation, glycosylation, ADP-ribosylation, sumoylation and ubiquitination, designated as histone codes that regulate gene expression epigenetically through various mechanisms [2]. These histone codes have been demonstrated to not only directly affect and change the structure of chromatin but also extensively involve the regulation of gene expression [3]. It has been revealed that the levels of histone methylation can be altered in abiotic and biotic stress responses of plants, and the methylation of some specific residues at the N-terminal tails of histone is closely related to the upregulated or downregulated expression of stress response genes [4]. Therefore, histone methylation has become one of the hot topics in epigenetic regulation research in recent years.

Various arginine (R) and lysine (K) residues at the N-terminal tails of histone can be methylated via protein arginine and lysine methyltransferases, respectively. Protein arginine methyltransferases (PRMTs) can methylate histone H3 at R2 (H3R2), R8 (H3R8), R17 (H3R17), R26 (H3R26) and H4 at R3 (H4R3) through transferring the methyl group from S-adenosylmeth ionine (Ado-Met) to the nitrogen atom of arginine side chain [5]. Generally, there are three main forms of methylated arginine: monomethylarginines (MMA), asymmetric dimethylarginines (ADMA) and symmetric dimethylarginines (SDMA). Based on various methylated arginine forms, PRMTs can be categorized into four major types, namely types I, II, III or IV enzymes [6,7]. The type I and II PRMTs can regulate gene expression through the methylation of histone tails. Among them, the type I PRMTs include PRMT1, PRMT2, PRMT3, PRMT4, PRMT6 and PRMT10 and usually methylate H3R2 and H4R3 residues to generate ADMA, leading to transcriptional activation and ribosomal biosynthesis [8]. In contrast, the type II PRMTs, including PRMT5 and PRMT9, are required for the formation of SDMA at H3R8 and H4R3 residues, resulting in transcriptional repression [9,10]. However, the type III PRMTs (mainly PRMT7) only catalyze the generation of MMA [11]. In addition, the type IV PRMTs can methylate secondary amine on arginine residues, which has only been revealed in yeast [11,12,13]. Since PRMTs are involved in regulating diverse biological processes in animals and yeast, their significance of PRMTs in the model plant *Arabidopsis* has been paid great attention to in recent years [14].

In plants, genes encoding PRMTs have been identified and analyzed in several species, including *Arabidopsis thaliana* [15], *Oryza sativa* [14], *Eucalyptus grandis* [16] and *Glycine max* [17]. Studies have revealed that PRMTs share rather conserved features in eukaryotic cells and play crucial roles in chromatin structure, RNA processing, altered gene transcription, transport and translation, DNA repair, cellular differentiation and signal transduction [18,19]. Previous studies have demonstrated that the absence of AtPRMT3 can lead to multiple developmental defects in *Arabidopsis*, including unbalanced polyribosome spectra and abnormal rRNA preprocessing, where rRNA precursor processing is required for ribosomal biogenesis [20]. In *Arabidopsis*, AtPRMT4a and AtPRMT4b, two orthologs of human PRMT4/CARM1 protein, can methylate histone H3R2, H3R17 and H3R26 in vitro and are required for the methylation at H3R17 in vivo. The double mutant of *AtPRMT4a* and *AtPRMT4b* genes exhibited an FLC-dependent late flowering phenotype [21]. In addition, AtPRMT5/Skb1 (Shk1binding protein 1) belongs to a type II PRMT and can result in the generation of SDMA at H4R3 residue in vitro, and the *atprmt5* mutant exhibited pleiotropic phenotypes, such as growth retardation, curly and dark green leaves and late flowering in FLC-dependent manner in *Arabidopsis* [22,23]. Increasing evidence has suggested that the PRMT5-mediated arginine methylation plays a crucial role in alterative splicing of normal pre-mRNA in plants and animals [24,25] and that the late-flowering phenotype shown in *atprmt5-1* and *atprmt5-2* mutants may be resulted from alterative splicing of flowering time regulatory genes associated with RNA processing [24]. Previous studies have demonstrated that PRMT5 acts as a key determinant of circadian period in *Arabidopsis* and Drosophila, which may link circadian cycle with alternative splicing [25]. It has also been revealed that the expression level of *FLOWERING LOCUS C (FLC)* gene is upregulated, which might be resulted from alternative splicing of *FLK*
*(Flowering Locus C)* in *atprmt5* mutant [26]. In addition, AtPRMT5/Skb1 has been revealed to regulate gene transcription and involve alterative splicing of pre-mRNA through symmetrically dimethylating H4R3 residues of histone and small nuclear ribonucleoprotein LSM4 and thereby confer a high salt stress tolerance in *Arabidopsis* [27]. Furthermore, AtPRMT10, a plant-specific type I PRMT, has been revealed to play divergent roles in flowering time control [15]. A mutation in the *AtPRMT10* gene resulted in late flowering by upregulating the transcript level of *FLC* gene. Moreover, the correlation between the *FLC* expression and flowering time and vernalization has been determined, which indicates that the *FLC* gene acts as an important determinant in natural variation of flowering time [28,29]. Further, the *Arabidopsis* FLC protein has been demonstrated to function as a flowering repressor, which may be involved in regulating related genes in autonomous or vernalization pathways. Previous studies have demonstrated that the *FLC* gene can function in repressing expression of the flowering regulatory genes *SOC1*
*(Super of overexpression COI)* and *FT*
*(Flowering Locus T)* through genetic and transgenic methods [30].

Furthermore, recent studies have revealed that many epigenetic factors are involved in various abiotic stress responses, and distinct chromatin modifications can be altered when plants are exposed to adverse environmental conditions, resulting in a dynamic chromatin environment to regulate gene expression [31]. At present, the cross-talk between diverse abiotic stress response pathways and epigenetic regulatory pathways has been thoroughly studied in plants [31]. However, the underlying mechanisms of epigenetic regulation of plant responses to heat stress remain to be elucidated, especially in regulating the dynamic histone arginine methylation patterns of stress-responsive genes, which partly depends on the catalytic function of PRMTs in plants. Moreover, proline is an essential multifunctional amino acid, which plays an important role in abiotic stress tolerance of plants. However, little is known about the biological functions of proline metabolism in plant responses to heat stress. Previous studies have revealed that short-term heat shock at 42 °C can result in proline accumulation in plant seedlings, and the external application of proline can induce the level of endogenous free proline and activities of antioxidant enzymes, followed by enhancing heat tolerance of plant seedlings [32]. Moreover, proline accumulation is mainly caused by the following three aspects: increasing the synthesis of proline, reducing the oxidation and degradation of proline and reducing the utilization of protein synthesis [33].

So far, much is well-known regarding proline metabolism in plants. The metabolic pathway of proline includes synthetic pathway and catabolic pathway. The synthetic pathway involves two enzymes, P5CS (pyrroline-5-carboxylate synthetase) and P5CR (P5C reductase). The degradation pathway involves other two enzymes, ProDH (proline dehydrogenase) and P5CDH (pyrroline-5-carboxylate dehydrogenase). [34]. Proline accumulation has been demonstrated to play adaptive roles in abiotic stress tolerance of plants [35]. Accumulated free proline could be implicated in adjusting cytosolic osmotic potential in order to save water [36], protect the membrane structure, sustain the structures of soluble proteins and activities of enzymes [37], act as reactive oxygen species (ROS) scavenger [38] and retain storage of carbon and nitrogen [39]. For example, overexpression of bean *P5CS1* gene in tobacco can slow down the decline of osmotic potential of transgenic plants under water stress [36].

Maize (*Zea mays* L.), as one of the most important crop species in the world and is vulnerable to environmental factors with climate change. Thus, how to use the methods of molecular biology to improve stress resistance and yield of maize is one of the hot topics of current biological research. Moreover, it has been demonstrated that plant *PRMT* genes are extensively involved in the regulation of growth and development and responses to various abiotic stresses. However, little is known regarding identification and function analysis of *PRMT* gene family in maize. In the present study, we carried out a comprehensive identification and functional analysis of *PRMT* genes in maize, including their phylogenetic relationships, gene and protein structures, conserved domain and motif architecture, chromosome location, gene duplication events and diverse expression profiles, which will facilitate further studies to unravel the exact biological roles of *ZmPRMT* genes in maize. Furthermore, the roles of *ZmPRMT1* in regulating flowering time and conferring heat stress tolerance were further clarified in transgenic *Arabidopsis*.

In this study, we selected transgenic *Arabidopsis* lines that overexpress maize *ZmPRMT1* gene to further reveal its functional roles in plants based on the conserved evolutionary relationship among ZmPRMT1 protein and some orthologous PRMT proteins in *Arabidopsis,* rice and sorghum. Although the direct use of *Arabidopsis* as a model plant for maize still has limitations, the current performance of *Arabidopsis* in plants is better than all other models. For example, when the purpose of the research is to gain a fundamental understanding of the specific growth process of plants, such as the regulation of flowering time, the *Arabidopsis* model system is still of reference significance [40]. Of course, it is also necessary to further verify the function of genes in maize through the overexpression and latest CRISPR/Cas9 gene editing techniques. In the case of flowering time regulation, the factors that advance the flowering time of *Arabidopsis* may inhibit the flowering time of rice (*Oryza sativa*) [41], and the network variation that affects the flowering time is significant even in gramineous plants [42,43]. Taken together, this study may contribute to an in-depth comprehension of the evolution of *ZmPRMT* genes and their crucial roles in maize growth, development and responses to abiotic stresses.

## 2. Results

### 2.1. Identification of the Members of PRMT Gene Family in Maize

To identify the total possible orthologs of *PRMT* gene family in maize, the PRMT protein sequences of *Arabidopsis*, rice and sorghum and their PRMT domains (Pfam: PF05185) were used as queries in the maize genome database. After the redundant sequences were removed, a total of eight maize PRMT proteins (namely ZmPRMT1–ZmPRMT8) were obtained. Further, these eight ZmPRMT sequences were scanned using Pfam and SMART databases to confirm the presence of the PRMT domain, respectively. The sequences of eight AtPRMTs, two OsPRMTs and two SbPRMTs were obtained from NCBI database for further investigation. The phylogenetic tree containing these representative AtPRMTs, OsPRMTs and SbPRMTs was constructed with ZmPRMT proteins, which was classified into three distinct subfamilies (Figure 1). Moreover, a total of eight *ZmPRMT* genes were uniformly named as *ZmPRMT1–ZmPRMT8* referring to their corresponding encoded protein in the above method section. All the conserved domains in PRMTs in maize were very similar to those PRMT proteins in *Arabidopsis*. Then the basic information such as the protein molecular weight, the isoelectric point and the number of amino acid residues was analyzed by ExPASY (https://web.expasy.org/protparam/) (accessed on 10 September 2021) (Table 1). The length of the *ZmPRMT* coding sequence varied from 798 bp from *ZmPRMT1* to 3201 bp for *ZmPRMT3*, with the respective coding potential of 306 and 1066 amino acids. The protein isoelectric point was between 5.27 (ZmPRMT4) and 8.36 (ZmPRMT3). Furthermore, the molecular weight varied from 29.86 KDa (ZmPRMT1) to 117.06 KDa (ZmPRMT3), indicating that ZmPRMT proteins share a large molecular weight range.

### 2.2. Phylogenetic and Conserved Domain Analysis of Maize PRMT Proteins

To investigate the phylogenetic relationships among PRMTs from maize, *Arabidopsis*, rice and sorghum, the multiple sequence alignment with MEGA7.0 was performed using the protein sequences of PRMTs from these plants and the unrooted phylogenetic tree was constructed from the alignment full-length protein sequences of eight AtPRMTs, two OsPRMTs, two SbPRMTs and eight ZmPRMTs through the Maximum-Likelihood method. Based on the comparison and analysis of evolutionary relationships among PRMT proteins, these proteins were categorized into three major subfamilies (Figure 1). In the first subfamily, maize ZmPRMT1 and ZmPRMT2 exhibit high homology with OsPRMT5, SbPRMT5 and AtPRMT5. In the second subfamily, maize ZmPRMT3, ZmPRMT4 and AtPRMT4A, AtPRMT4B belong to orthologous proteins. In the third subfamily, ZmPRMT7, OsPRMT10, AtPRMT10 and SbPRMT10 belong to orthologous proteins, and ZmPRMT5, ZmPRMT6 and ZmPRMT8 share high homology with AtPRMT1A, AtPRMT3, AtPRMT6 and AtPRMT11. Further, the genetic structure of maize *PRMT* genes is highly conserved, and the structure is simpler than that of other plants *PRMTs* (Figure 1a), which is consistent with their motif distribution (Figure 1b).

In order to understand the difference of the domain architecture, the SMART database was used to determine the structure types of eight maize PRMT proteins with default parameters (Figure 1c). According to their relationships with *Arabidopsis*, rice and sorghum PRMT proteins, the eight ZmPRMT proteins were classified into three different subfamilies, including the subfamily I (two members), the subfamily II (two members) and the subfamily III (four members). In subfamily I, ZmPRMT1 and ZmPRMT2 share two conserved domains including PRMT5 (PF05185) and PRMT5_C. The PRMT5 domain is the conserved functional domain of arginine methyltransferase. ZmPRMT2 has a complete PRMT_Tim domain at the N-terminal, which may play a role in the quaternary structure of this protein. In subfamily II, ZmPRMT4 has only one conserved PrmA domain, while ZmPRMT3 has another REF domain at the C-terminal. Among the members of subfamily III, all these proteins share several common conserved domains: MTS, PrmA, Methyltransf_11 and Methyltransf_25. Among them, ZmPRMT5, ZmPRMT6 and ZmPRMT7 contain more than six domains, which are concentrated in the anterior middle segment of the protein sequence, and ZmPRMT5 and ZmPRMT7 have only one Met_10 domain at the C-terminal. The conserved domains of ZmPRMT8 are concentrated at the N-terminal. In conclusion, the distribution of conserved domains of each protein is related to the evolution of each other.

### 2.3. Chromosomal Localization and Gene Duplication of PRMT Gene Family in Maize

The physical locations of *ZmPRMT* genes on chromosomes were investigated to generate the chromosomal position graphics of *ZmPRMT* genes. The graphics showed that all eight *ZmPRMT* genes were distributed unevenly across seven of all the ten chromosomes in the maize genome (Figure 2). These newly identified genes were distributed individually in various regions of these chromosomes (i.e., telomere, near centromere and other regions). Chromosome 7 has the highest number of *ZmPRMT* genes, i.e., two, while Chromosomes 1, 2, 4, 5, 6 and 10 have only one in each, respectively. In detail, *ZmPRMT3* and *ZmPRMT8* were located on Chromosome 7, whereas the *ZmPRMT6*, *ZmPRMT4*, *ZmPRMT2*, *ZmPRMT1*, *ZmPRMT7* and *ZmPRMT5* were located on Chromosomes 1, 2, 4, 5, 6 and 10, respectively. Furthermore, *ZmPRMT2*, *ZmPRMT3* and *ZmPRMT5* are distributed near the telomeres of chromosomes. Considering the importance of telomere structure for maintaining chromosome stability and ensuring the complete replication of genes on chromosomes, the above three genes are considered to be very conserved in the process of species evolution and may play an irreplaceable role in normal growth and developmental process of maize. Furthermore, gene duplication events were investigated to determine the evolutionary patterns of the maize *PRMT* gene family. Based on the analysis of sequence alignment, it was revealed that a pair of genes (*ZmPRMT3*/*ZmPRMT4*) was involved in the segmental duplication of maize for they share 88.67% homology in the sequences (Appendix A**)**. This result indicates that gene duplication events may have occurred during the evolution process of *PRMT* genes to preserve the function of PRMT proteins.

### 2.4. The Cis-Acting Regulatory Elements in the Promoter of Maize PRMT Genes

To further investigate the potential regulatory mechanism in biotic or abiotic stress responses, the cis-acting elements were detected in 2000 bp upstream of promoter of *ZmPRMT* genes via using the PlantCARE database. All of the 16 cis-acting regulatory elements associated with stress and hormones were detected in the promoter regions of the *PRMT* gene in maize (Figure 3). Most *ZmPRMT* genes contain ARE that is related to the anaerobic reaction, except *ZmPRMT6*. Further, most *ZmPRMT* genes share the CGTCA-motif (MeJA reactive element), TGACG-motif (MeJA reactive element), ABRE (abscisic acid reactive element) and LTR (low-temperature relative element). The results indicated that these genes not only respond to hormone but also may respond to abiotic stress. In addition, MBS (drought induction element), TC-rich repeats (defense and stress response element), GARE motif (gibberellin response element), P-box (gibberellin response element), TGA element (auxin response element) and TCA element (salicylic acid response element) were detected in the 2000 bp upstream region of the promoter of these genes. Taken together, these results revealed that expression of maize *PRMT* genes could be modulated by various hormones and adversity stress, which may participate in the regulation of biotic or abiotic stress responses and hormone signal transduction.

### 2.5. Analysis of Microarray Expression Profile of Maize PRMT Genes in Different Tissues

To further explore the transcription patterns of maize *PRMT* genes, the expression profile of eight *ZmPRMT* genes in 60 different developmental periods was characterized with microarray data (Figure 4). Heat map shows that *ZmPRMT* genes have a trend of differential expression in different growth periods of all these tissues. The signal values for all these *ZmPRMT* genes were shown in Appendix A. According to the heat map, some specifically expressed *ZmPRMT* genes in tissues or organs were discovered at 60 diverse developmental stages. All eight *ZmPRMT* transcripts investigated were expressed in the whole growth process, although these members were expressed at low levels in some tissues at different growth and development stages. In addition, the results showed that some *ZmPRMT* genes begin to express at a certain time during plant growth and development, indicating that these *ZmPRMT* genes are very important in maize growth and development processes. Moreover, we found that the expression patterns of *ZmPRMT* genes can be roughly classified into two periods with the R2_Outer Husk and the R2_Innermost Husk as the boundaries. In the first period, the expression levels of most *ZmPRMT* genes were generally very low at various tissues. However, some *ZmPRMT* genes were highly expressed in some specific tissues such as 6DAS_GH_WTeoptile, 6DAS_GH_Primary Root, VE_Primary Root, V1_Stem and SAM, V3_Stem and SAM, V13_Immature Tassel and V18_Immature Cob, implying that these genes might play important roles in these specific tissues at different periods of maize growth process. Furthermore, according to the heat map, several *ZmPRMT* genes such as *ZmPRMT2*, *ZmPRMT3* and *ZmPRMT4* were also revealed to be highly or specifically expressed in some tissues. In the second period, the majority of *ZmPRMT* genes were expressed highly or specifically at different developmental stages of seed and endosperm after pollination. To summarize, the expression pattern showed that the transcriptional levels of *ZmPRMT* genes in seed and endosperm were obviously higher than those in roots, stems and leaves. Taken together, the results demonstrated that these identified *ZmPRMT* genes exhibited differential expression patterns at diverse growth and development stages of maize, suggesting that these genes may function in multiple tissues.

### 2.6. Expression Profile Analysis of Maize PRMT Genes under Abiotic Stress Treatment

To further confirm the responsiveness of *ZmPRMT* genes to abiotic stresses, the expression profile of these eight genes was further explored by qRT-PCR analysis and with at least three biological repeats to make sure the reliability of the qRT-PCR results. The experiment was accomplished by using the cDNA of leaves, stems and roots of maize seedlings, which were exposed to three diverse abiotic stress treatments, including heat, drought and salt. Firstly, the different maize tissues (including leaves, stems and roots) at five different time points (0 h, 1 h, 2 h, 4 h and 8 h) under heat stress treatment were collected to investigate the transcription levels of these eight *ZmPRMT* genes by using qRT-PCR, respectively (Figure 5). Firstly, the expression levels of these eight *ZmPRMT* genes in maize leaves, stems and roots at five different time points (0 h, 1 h, 2 h, 4 h and 8 h) after heat treatment were detected by qRT-PCR analysis. After heat treatment, these eight *ZmPRMT* genes displayed differential accumulation of expression levels. One of the most obvious results we can observe is that the majority of *ZmPRMT* genes exhibited differential upregulated expression levels in leaves after heat treatment, whereas only one *ZmPRMT* gene (*ZmPRMT5)* in stems and two *ZmPRMT* genes (*ZmPRMT1, ZmPRMT5)* in roots showed the significantly upregulated expression levels after heat treatment compared with those in leaves. Only two *ZmPRMT* genes (*ZmPRMT4* and *ZmPRMT6)* in leaves didn’t show upregulation after heat stress at different time points, while the others exhibited upregulation in leaves at distinct time hours, indicating that most of these *ZmPRMT* genes may play potential roles in heat stress responses. Moreover, under heat treatment, the majority of these *ZmPRMT* genes showed more obvious responses in leaves than those in stems or roots. Based on the above analysis, it can be speculated that the *ZmPRMT* genes in leaves might be involved in heat stress responses of plants and more sensitive than those in steams and roots. Therefore, these results indicate that plant leaves rather than stems and roots may be the main organs of plants in heat stress responses. Moreover, it is noticeable that the upregulated expression patterns of *ZmPRMT1* and *ZmPRMT5* in all the three different tissues revealed that these two genes might play crucial roles in maize in response to heat stress.

Further, we performed the drought stress treatment. The majority of *ZmPRMT* genes showed differential downregulated expression levels in leaves after drought treatments, whereas the most of *ZmPRMT* genes in stems and roots showed significantly upregulated expression levels after drought treatments compared with those under normal conditions (CK) (Figure 6). For example, there were only two *ZmPRMT* genes (*ZmPRMT5* and *ZmPRMT6*) in leaves exhibiting upregulation under drought treatment, whereas only two *ZmPRMT* genes (*ZmPRMT1* and *ZmPRMT2*) in the stems and only one *ZmPRMT* gene (*ZmPRMT6*) in the roots showed lower expression levels under drought treatment. It should be noted that the expression profiles of these *ZmPRMT* genes in maize under drought treatment were different from those under heat treatment. Especially in roots, most of the *ZmPRMT* genes expressed highly after drought treatment, and most of them were upregulated significantly and responded rapidly at 1 h after drought treatment in roots, indicating that these genes showed faster and stronger responses in roots than those in stems under drought stress. Similarly, it is also possible that the plant stems and roots rather than leaves are the main organs of plants in response to drought stress.

Finally, the salt stress treatment was completed. The majority of *ZmPRMT* genes showed differential downregulated expression patterns in all three different tissues after salt treatment, and there were only one *ZmPRMT* gene (*ZmPRMT1*) in stems and three *ZmPRMT* genes (*ZmPRMT1*, *ZmPRMT2* and *ZmPRMT4*) in roots exhibited higher expression levels under salt treatment than that under normal conditions (CK) (Figure 7). Therefore, these results show that salt stress can significantly inhibit the expression of these genes at all time points. Taken together, these results suggest that *ZmPRMT* genes may play an important role in responding abiotic stress and most of them exhibited immediate response to abiotic stress.

### 2.7. Generation of Overexpressed ZmPRMT1 Transgenic Arabidopsis Plants

To evaluate the function of *ZmPRMT* genes, we firstly constructed the overexpression vector of *ZmPRMT1* gene under 35 s promoter through genetic engineering method. The *ZmPRMT1* clone map and the double enzyme (Hind3/Xba1) digestion map of the recombinant plasmid were shown in Appendix A and S2B, respectively. Then, the recombinant vector with the *ZmPRMT1* gene was transformed into the *Arabidopsis* line using *Agrobacterium*i-mediated method by dipping the *Arabidopsis* floral to obtain the overexpressed *Arabidopsis* plants. Consequently, a total of 14 transgenic lines were generated in this study (Appendix A). Among them, three representative stable homozygous lines 2, 6 and 7 were further selected for functional analysis. The expression of *ZmPRMT1* gene in three transgenic *Arabidopsis* lines was validated through qRT-PCR. In addition, according to the analysis of phylogenetic tree, the *ZmPRMT1* gene was located in the Group I and its orthologous gene (*AtPRMT5*) (Appendix A) has been confirmed to regulate flowering time in *Arabidopsis*. Thus, the *ZmPRMT1* gene may share similar functions with the orthologous *AtPRMT5* gene, by which the protein encoded encompasses high homology, locates in nucleus and functions in repressing target genes by methylating H4R3 and H3R8 residues of histone, as well as transcription factors/regulators. Thus, this result implies that the *ZmPRMT1* gene may be involved in the modulation of vegetative growth and control of flowering time in plants.

### 2.8. Overexpression of ZmPRMT1 Gene Advances the Flowering Time of Transgenic Arabidopsis

Previous studies have demonstrated that *AtPRMT5* in *Arabidopsis* plays a critical role in regulating flowering time by virtue of FLC determinative factor [30], which indicates that the *ZmPRMT1* gene, the orthologous gene of *AtPRMT5* in maize, may have similar physiological function to *AtPRMT5*—that is, to advance the flowering time of plants in an FLC-dependent manner. Therefore, in this study, *ZmPRMT1* gene was selected from the *ZmPRMT* gene family to explore its expression pattern and genetic interaction with some flowering-related regulatory genes in transgenic lines. Firstly, the phenotype of these four different lines including WT, transgenic lines 2, 6 and 7 were surveyed in growth progress around flowering. These results suggest that there is no obvious difference between the transgenic *Arabidopsis* and WT lines in seedlings. However, the transgenic *Arabidopsis* lines were flowered earlier than WT plants, as well as the leaf numbers of them were slightly lower than those of WT plants when these *Arabidopsis* plants have been cultivated for about 30 days (Figure 8a,b), which suggested that overexpression of *ZmPRMT1* gene might result in early flowering in transgenic *Arabidopsis* through advancing the flowering time of these plants.

To further confirm the potential regulatory pathways of overexpressed *ZmPRMT1* gene in regulating flowering time of transgenic *Arabidopsis*, the *Arabidopsis* flowering-related regulatory genes, including *FLC*, *SOC1*, *FT*, MADS Affecting Flowering genes (*MAF1*, *MAF2*, *MAF3*, *MAF4* and *MAF5*) were investigated through qRT-PCR, respectively (Figure 9). The experiment was repeated biologically for at least three times to ensure the accuracy of qRT-PCR analysis. The *Arabidopsis FLC* gene has been reported to encode a MADS domain protein, which acts as an inhibitive factor of flowering promoting genes: *SOC1* and *FT* [28,44,45,46,47]. The results showed that the *ZmPRMT1* gene exhibited differential upregulated expression levels in all three transgenic lines compared with those in WT after flowering. The expression levels of *FLC* in transgenic and WT lines were significantly downregulated after flowering, whereas the expression levels of *FT* and *SOC1* in all four lines were upregulated after flowering. The *SOC1* gene in transgenic line 2 was shown to exhibit higher expression level than two other transgenic lines and WT after flowering. The *FT* gene was significantly upregulated in transgenic lines 2, 6 and 7 than that in WT after flowering. Furthermore, the expression levels of *MAF* genes, which shared highly conversed MADS domain with *FLC*, were also be investigated in these *Arabidopsis* lines after flowering. The transcript levels of *MAF* genes (*MAF1-4*) showed a consistent decrease with the *FLC* expression level after flowering in all investigated *Arabidopsis* lines, whereas *MAF5* exhibited high transcript level after flowering in all investigated *Arabidopsis* lines. Taken together, the early flowering phenotype of these transgenic *Arabidopsis* may be often closely linked to the expression of *ZmPRMT1*. Furthermore, *ZmPRMT1* gene may share the similar physiological function as *AtPRMT5* to promote plant flowering time in FLC-dependent manner. Given the expression pattern of flowering-related regulatory genes, it can be speculated that *ZmPRMT1* gene may be required for the promotion of flowering in maize by modulating the expression of flowering-related regulatory genes under natural environmental conditions.

### 2.9. Overexpression of ZmPRMT1 Gene Enhances Heat Tolerance in Transgenic Arabidopsis

To further explore heat stress-responsiveness of *ZmPRMT1* gene in transgenic *Arabidopsis*, the leaves were sampled at two different time points (0 h and 8 h) under heat stress treatment and performed for the qRT-PCR analysis, respectively. The experiment was repeated biologically at least three times to ensure the accuracy of qRT-PCR analysis. Firstly, the phenotype of these four different lines: WT, transgenic *Arabidopsis* lines 2, 6 and 7 were surveyed under 0 h and 8 h heat stress. Under the control condition, no significant phenotypic difference was observed between transgenic *Arabidopsis* and WT lines. However, under heat stress condition (42 °C), the transgenic *Arabidopsis* lines 2, 6 and 7 were respectively more resistant to heat stress than WT. Especially, the transgenic *Arabidopsis* lines exhibited higher vitality than the WT plants (Figure 10). Overall, these results indicate that the overexpressed *ZmPRMT1* gene in *Arabidopsis* can increase thermotolerance.

Furthermore, to explore molecular mechanism of heat tolerance of transgenic *Arabidopsis* in response to heat stress, we compared the proline contents of WT and transgenic *Arabidopsis* lines under 42 °C treatment at 0 h and 8 h. The proline contents of these 4 *Arabidopsis* lines exposed to heat stress treatment have been displayed to increase significantly compared with those under the normal conditions (Figure 11). For example, the proline contents in the transgenic *Arabidopsis* lines 2 and 6 were revealed to exhibit increased accumulation compared with that in WT plants after heat treatment, indicating that the *ZmPRMT1* gene could be involved in modulating heat-induced proline accumulation in transgenic *Arabidopsis*. To further explore heat stress tolerance mechanism of *ZmPRMT1* gene in proline metabolic pathway, the *Arabidopsis* leaves were further sampled after heat treatments at 0 h and 8 h and carried out for the qRT-PCR analyses to further discern the transcription patterns of some genes associated with proline metabolic pathway (Figure 11). The results showed that the related genes of proline synthesis pathway showed differential upregulated expression levels, whereas the related genes of proline degradation pathway exhibited differential downregulated expression levels after heat treatment. The *P5CS1* and *P5CR* genes in proline synthesis pathway showed significantly upregulated expression levels in all three different transgenic *Arabidopsis* lines compared with in WT plants after heat treatment, while the *P5CDH* and *PDH* genes of proline degradation pathway showed significantly downregulated expression levels in all three different over-expression lines compared with in WT plants under heat stress. Taken together, these results reveal that the overexpression of *ZmPRMT1* gene can result in greater proline accumulation by increasing the expression levels of proline synthesis pathway genes and decreasing the expression levels of proline degradation pathway genes, which is likely to lead to enhance of water potential and osmotic potential necessary for holding photosynthetic activity to alleviate heat stress and thereby enhance the heat stress tolerance of transgenic *Arabidopsis*.

## 3. Discussion

In plants, histone methylation is an important epigenetic modification involved in various biological processes by adjusting the homeostasis of histone methylation and demethylation [48,49]. Histone methyltransferase and demethylase are indispensable of accommodating the homeostasis of histone methylation in growth, development and responses to biotic and abiotic stresses in plants. Protein arginine methyltransferase (PRMT) can regulate the methylation status of arginine residues at histone tails and thereby activate or repress the transcription of target genes, which may play vital roles in plant growth and development and responses to abiotic stresses. In this study, a comprehensive set of eight nonredundant *ZmPRMT* genes were identified and analyzed from the AGPv4 version of maize (B73 inbred line) genome, including their phylogenetic relationships, gene and protein structures, conserved domain and motif architecture, chromosome location, gene duplication events and diverse expression profiles in maize developmental process and responses to various abiotic stresses. In addition, we also explored the effect of *ZmPRMT1* gene on heat stress and flowering time control via transgenic *Arabidopsis*.

Firstly, the analysis of phylogenetic relationship was carried out to provide insights into the evolution of *PRMT* gene family members and gene multiplicity in maize. In the present study, an unrooted phylogenetic tree was constructed through multiple sequence alignments of conserved PRMT domain-containing proteins from these putative PRMT proteins in maize and their representative orthologs from *Arabidopsis*, rice and sorghum. In our investigation and exploration of PRMT proteins, we discovered that maize *PRMT* genes were mainly clustered into three distinct subfamilies. In addition, the type IV PRMT has only been discovered in yeast so far [11,12,13]. Furthermore, although maize possesses a larger genome size (2300 Mbp) compared with *Arabidopsis* (125 Mbp) and rice (389 Mbp), only eight *PRMT* genes were identified and analyzed in maize. The number of *ZmPRMT* genes is similar to that in *Arabidopsis* (nine) or rice (eight). The unique phenomenon may be resulted from the occurrence of less gene duplication events in maize genome or undergoing a large gene loss during the process of maize genome duplication.

Furthermore, the analysis of conserved domain and motif architecture revealed evolutionarily conserved structure features and highly similarity among members of PRMT protein family, indicating that the PRMT proteins are highly conserved during the process of evolution. In addition, the gene structure map further reveals that the position and phase of intron/exon in the same group or subgroup exhibit considerable difference between maize and other plants, which may be resulted from too many nonfunctional sequences in the huge genome of maize. The genetic structure of maize *PRMT* genes is simpler than that of *PRMT* genes in other plants, but they share similar motif or domain distribution, implying that the protein structure in the same group or subgroup may be quite conserved during the process of evolution. For example, all of the ZmPRMT proteins investigated are mainly clustered into three distinct subclasses and each ZmPRMT protein contains the arginine methyltransferase-related conserved domains. It is noteworthy that the ZmPRMT1 and ZmPRMT2 proteins share two conserved domains, including PRMT5 (PF05185) and PRMT5_C, which act as the functional domain of arginine methyltransferase. In addition, the ZmPRMT2 protein has a complete PRMT_Tim domain at the N-terminal, which may play a crucial role in the quaternary structure of protein. In the subfamily II, ZmPRMT4 has only one conserved PrmA domain, while ZmPRMT3 has another REF domain at the C-terminal. Among the members of subfamily III, the majority of these proteins share several common conserved domains: MTS, PrmA, Methyltransf_11 and Methyltransf_25. Other proteins have different conserved arginine methyltransferase-related domains in different positions. In conclusion, the distribution of conserved domains of each protein may be closely linked to the evolutionary functions of each other. For example, it has been reported that the PrmA domain from *Arabidopsis* is dually targeted to chloroplasts and mitochondria. In addition, the conserved PrmA domain in photosynthetic eukaryotes accompanied by the methylated binding sites of translation factor to ribosome indicates that the PrmA domain in plants may participate in binding extra post-translational modifications or enhancing the function of ribosome [50]. Moreover, the REF domain (Apurinic/apyrimidinic endonuclease/redox factor) contained in ZmPRMT3 has been revealed to function in cellular responses to DNA damage or oxidative stress [51].

To further confirm potential functions of *ZmPRMT* genes in developmental processes and stress-responsiveness of them to abiotic stresses, the expression patterns were conducted by applying the available transcriptomic data at different developmental stages of maize [52]. The result demonstrated the *ZmPRMT* genes showed apparent differential expression patterns in the main tissues of different growth periods, indicating that these genes may function in growth and development processes of maize. Moreover, among 60 developmental periods, some genes were revealed to express in a tissue-specific manner, whereas the other genes were shown to express in a time-specific manner. For example, if we had divided the expression profile of these *PRMT* genes into two stages, we could found that some tissue/organ specific genes such as *ZmPRMT2*, *ZmPRMT3* and *ZmPRMT4* were revealed to express highly or specifically in some tissues of the first stage such as 6DAS_GH_WTeoptile, 6DAS_GH_Primary Root, VE_Primary Root, V1_Stem and SAM, V3_Stem and SAM, V13_Immature Tassel and V18_Immature Cob, whereas the time-specific genes like *ZmPRMT1* were expressed highly in the whole second stage. Overall, the majority of *ZmPRMT* genes show obviously differential expression levels, suggesting that *ZmPRMT* genes may play crucial roles in the growth and development of maize.

Furthermore, previous studies have showed that plant PRMTs participate extensively in diverse regulation of transcriptional and post-transcriptional levels, which are involved in gene expression, mRNA processing, translation and intracellular signaling during the growth and development of plants [53]. For example, the PRMT1 can mediate asymmetric dimethylation of histone H4 arginine 3 (H4R3me2a), as well as a number of nonhistone proteins, such as C/EBPα12, Twist113 and Gli114, and thereby promote transcription [54]. Moreover, it has been reported that EgPRMT1 plays a critical role in the initiation and elongation of root hair, which is resulted from the methylation of β-tubulin that is associated with cytoskeleton formation [16]. Also, AtPRMT3 has been revealed to participate in RNA processing and ribosomal biogenesis in *Arabidopsis*. Moreover, AtPRMT5 belongs to the Type II PRMT and is extensively involved in pre-mRNA splicing [25], flowering time [24], salt stress tolerance [27], primary root length [22], root stem cell maintenance during DNA damage [26] and circadian rhythms [26]. In this study, to confirm the expression patterns and their stress-responsiveness of these newly-identified *ZmPRMT* genes in response to various stresses including heat, drought and salt treatments in three different tissues, the expression patterns of *ZmPRMTs* under three abiotic stresses were performed by means of qRT-PCR analysis in three different tissues. The results demonstrated that all *ZmPRMT* genes showed obviously differential expression levels in three distinct tissues under heat, drought and salt stress treatments. For instance, the majority of *ZmPRMT* genes were significantly upregulated in leaves at different time points after heat treatment, whereas most of *ZmPRMT* genes were downregulated in stems and roots after heat treatment. After drought treatment, the majority of *ZmPRMT* genes showed differential downregulated expression levels in leaves, whereas most of *ZmPRMT* genes in stems and roots showed significantly upregulated expression levels. Furthermore, the majority of *ZmPRMT* genes showed differential downregulated expression levels in all three tissues after salt treatment, and there was only 1 *ZmPRMT* gene (*ZmPRMT1*) in stems and three *ZmPRMT* genes (*ZmPRMT1*, *ZmPRMT2* and *ZmPRMT4*) in roots exhibited upregulated expression levels under salt treatment. Based on the above analysis, these results suggest that the majority of *ZmPRMT* genes may play potential roles in response to diverse abiotic stresses. One possible explanation is that this phenomenon may be resulted from the specifically temporal and spatial expression regulation when responding to abiotic stresses. Additionally, the promoter analysis further demonstrated that many stress- and hormone-related cis-elements in promoter regions of *ZmPRMT* genes might be involved in transcriptional regulation of diverse abiotic stress responses.

In plants, the timing of transition from vegetative to reproductive development has been revealed to become a critical adaptive trait, which is necessary for plants to accomplish flower development, pollination, and seed production in favorable conditions. Previous studies have showed that the continuous growth under low temperature conditions can accelerate flowering in most plant species [45]. This phenomenon is referred to vernalization, which is a key determinant involving the switch from vegetative to reproductive development. In *Arabidopsis*, the *FLOWERING LOCUS C (FLC)* is required for most vernalization-requiring *Arabidopsis* accessions transcriptionally. The *Arabidopsis FLC* gene has been revealed to encode a MADS domain protein that functions both in leaves and in the apical meristem as a repressor of flowering promoting genes: *SOC1* and *FT* [15,28,30,46,47]. Similar to *FLC*, the expression levels of *MAF* genes are usually dependent on vernalization regulation. Vernalization represses the expression of *MAF1*, *MAF2*, and *MAF3*, but it induces the expression of *MAF5* and does not significantly affect the expression of *MAF4* [55]. According to the previous studies, *AtPRMT5* and *AtPRMT10* showed critical effects on flowering time and *FLC* mRNA levels. In this study, *ZmPRMT1*, an orthologous gene of *AtPRMT5*, was assumed to function in regulating flowering time in plants. Thus, to further explore the role of *ZmPRMT1* in flowering time regulation, we constructed transgenic lines overexpressing *ZmPRMT1* gene in *Arabidopsis* for preliminary functional verification. Firstly, the results of phenotypic analyses revealed that the most common effect of transgenic *Arabidopsis* overexpressing *ZmPRMT1* was early flowering. Then, the flowering-related regulatory genes including *FLC*, *SOC1*, *FT*, and *MAF1* to *MAF5* were investigated through quantitative real-time PCR analysis, respectively. Consistent with our observations, the expression levels of *FLC* in transgenic *Arabidopsis* lines 2 and 6 decreased compared with that in WT plants after flowering, whereas the expression levels of *FT* and *SOC1* in transgenic *Arabidopsis* lines 2, 6 and 7 increased after flowering. In addition, the expression levels of *MAF1-4* genes showed a consistent decrease with the *FLC* expression level after flowering in all investigated *Arabidopsis* lines, whereas *MAF5* exhibited high transcript level after flowering in all investigated *Arabidopsis* lines. Furthermore, previous studies showed that some other *FLC* clade members are required for FLC protein binding to the chromatin associated with the *FT* and *SOC1* genes, indicating that these proteins may be involved in regulating flowering via MADS-domain complexes [56].

Therefore, we conclude that *ZmPRMT1* is required for the promotion of flowering in plants, which may control the floral transition in an FLC-dependent manner based on the above analysis. Moreover, histone arginine methylation has been elucidated to be a conserved epigenetic mechanism involving dynamics regulation of eukaryotic chromatin in three different methylation manners. Among them, monomethylation (MMA) and asymmetric dimethylation (ADMA) are generally involved in transcriptional activation, whereas symmetric dimethylation (SDMA) is associated with transcriptional silencing [9]. For instance, *Arabidopsis PRMT5* can symmetrically methylate some arginine residues of relative proteins involving RNA processing and histones, specifically histone 4 [10]. In this study, it can be speculated that *ZmPRMT1* might promote flowering time by repressing *FLC* expression level, as well as regulating other related *MAF* genes. Therefore, a schematic model of *ZmPRMT1*-mediated flowering time regulation in transgenic *Arabidopsis* was proposed in this study (Figure 12).

Plants are sessile organisms that are subject to constantly endure a variety of adverse environmental conditions, some of which can result in abiotic stress responses. High temperatures causing heat stress responses generally bring about the disturbance of cellular homeostasis and the impedance of growth and development in plants. One of the major stresses in crop plants is heat stress, which is usually accompanied by other stresses that are resulted from extra environmental conditions such as drought or salinity [57]. Accumulating studies have revealed that proline, as an essential multifunctional amino acid, as well as a stress signal, is extensively involved in multiple physiological pathways such as adjusting osmotic potential, scavenging reactive oxygen species and buffering redox reactions, and functions as a small molecular chaperone, as well as a plant development signal [58,59,60,61]. Moreover, previous studies have also revealed that short-term heat shock at 42 °C can result in proline accumulation in maize seedling and the exogenous application of proline can improve the level of endogenous free proline and thereby enhance the activities of antioxidant enzymes, followed by an increased heat tolerance of maize seedling [32]. Thus, we proposed a schematic model of the *ZmPRMT1* gene involved in proline catabolism for improving heat tolerance in transgenic *Arabidopsis* (Figure 13). In this study, our results demonstrated that the proline accumulation of WT and transgenic *Arabidopsis* lines under heat stress treatment all increased. However, the proline accumulation in transgenic *Arabidopsis* lines increased more significantly than that in WT plants. Furthermore, the analysis of qRT-PCR showed that *P5CS1* and *P5CR* gene in proline synthesis pathway exhibited upregulated expression levels in transgenic *Arabidopsis* lines compared with in WT plants after heat stress treatment, whereas *P5CDH* and *PDH* gene in proline degradation pathway exhibited significantly downregulated expression levels in transgenic *Arabidopsis* lines compared with in WT plants under heat stress treatment. Based on previous studies, *Arabidopsis AtPRMT5* usually functions in repressing target genes by symmetrically demethylating histone H4 Arginine 3 (H4R3me2s) in vitro [22]. If *ZmPRMT1* is involved in regulating proline-related genes directly in this study, it might be expected to reduce the expression levels of proline-related regulatory genes including *P5CS1*, *P5CR*, *P5CDH* and *PDH* in transgenic *Arabidopsis*. However, overexpression of the *ZmPRMT1* gene in transgenic *Arabidopsis* lines was revealed to result in increased expression levels of *P5CS1* and *P5CR* genes, as well as decreased expression levels of *P5CDH* and *PDH* genes. Together with these results, it can be speculated that the protein encoded by *ZmPRMT1* may be involved in both positive and negative regulatory effects on gene expression through modifying diverse histone methylation of specific arginine residues. Overall, the results suggested that the maize *ZmPRMT1* gene might play a crucial role in resisting heat stress response in plants. However, the underlying regulatory mechanism of this gene still remains to be further elucidated.

## 4. Materials and Methods

In this study, a comprehensive strategy combining bioinformatics and expression profiling analysis were used to identify all *ZmPRMT* genes and explore their function in response to abiotic stress. Furthermore, through the *Arabidopsis* model overexpressing maize *ZmPRMT1* gene, the molecular mechanism of *ZmPRMT1* gene responding to heat stress and promoting early flowering time was clarified in this study. The schematic flowchart of the study is shown in Figure 14.

### 4.1. Plant Materials and Growth Conditions

The plants of maize B73 inbred line were grown in the temperature incubator at 28 °C under long-day conditions (15 h light and 9 h dark) and 60% relative humidity. In order to explore the expression pattern of maize *ZmPRMT* genes under three abiotic stresses (heat, drought and salt), we set up three stress treatment methods. At 21 days after emergence, heat stress was induced by 42 °C. The maize seedlings grown in the dark incubator at 28 °C with enough water were used as the control. Maize seedlings in the same growth period were used as treat materials under salt stress. The seedlings of the treat group were cultured in the nutrient solution containing 200 mM NaCI, while the seedlings of the control group were grown in the nutrient solution lacking NaCI. After being treated with NaCI, the maize seedlings were washed with distilled water for subsequent experiments. In the drought treatment, the three-week-old seedlings were gently pulled out of the soil and placed on clean white paper in a dark incubator as the treat group, while the seedlings of the control group were grown under normal conditions. The root, stem and leaf tissues of maize seedlings were collected at 0 h, 1 h, 2 h, 4 h and 8 h treated with heat, salt and drought stress and stored in liquid nitrogen to extract RNA.

### 4.2. Identification of the Members of PRMT Gene Family in Maize

To identify all possible orthologs of *PRMT* gene family in maize, we took the *PRMT* genes in *Arabidopsis*, rice and sorghum as homologous reference genes and carried out the following analysis. The whole-genome sequences of maize were downloaded from the maize genomic database (http://www.maizesequence.org/index.html) (accessed on 5 September 2021) by referring to our previous studies [49]. The BLASTP program (*p*-value < 1 × 10^−5^) was used to confirm proteins containing PRMT domain from the maize genomic database using the published eight *Arabidopsis* PRMTs (AtPRMTs), two rice PRMTs (OsPRMTs) and two sorghum PRMTs (SbPRMTs), with sequences of their proteins and PRMT domains as queries, respectively. The information of PRMT genes in these plants were listed in Appendix A. The Hidden Markov Model (HMM) program was used to identify putative PRMT protein sequences in maize genome database, which was downloaded from the SMART database (http://smart.embl-heidelberg.de/) (accessed on 3 September 2021) and Pfam database (http://pfam.xfam.org/) (accessed on 3 September 2021) [62,63]. Redundant protein sequences were removed manually by searching against the SMART database and the NCBI database (https://www.ncbi.nlm.nih.gov) (accessed on 3 September 2021). The newly identified genes were assigned by referring to the subfamily classification order of this gene family, combined with their phylogenetic relatedness to orthologous proteins of PRMTs in *Arabidopsis*, rice and sorghum. Then, the genome-wide files of maize B73 genome were downloaded to obtain the basic information of each *PRMT* gene from the maize genomic databases, including the length of CDS, the number of amino acids and the location of chromosomes. The physicochemical parameters of these putative proteins, including molecular weight (kDa) and isoelectric point (pI), were calculated using the online calculation pI/Mw tool ExPASy (http://www.expasy.org/tools/) (accessed on 10 September 2021), and the parameter was set to average [64]. Other relevant information was collected from the NCBI database. Similarly, the protein sequences of corresponding orthologs in *Arabidopsis*, rice and sorghum were also retrieved, and a dataset was created for bioinformatics analysis.

### 4.3. Phylogenetic Relationship, Gene Structure and Conserved Domain Analyses of the PRMT Gene Family in Maize

To clarify the phylogenetic relationship and structural characteristics among *PRMT* genes, we constructed an evolutionary tree, a gene structure map and a conservative structure map. All predicted PRMT protein sequences of maize and their orthologous sequences from *Arabidopsis*, rice and sorghum were aligned by Clustal-W software using default parameters [65,66]. All protein sequences were downloaded from the NCBI database. Then, MEGA7.0 program was used to analyze the phylogenetic relationship among these different species via the Maximum-Likelihood method with default parameters. The genetic structures of maize *PRMT* genes were obtained using online website GSDS (http://gsds.gao-lab.org/) (accessed on 12 September 2021), according to the alignment of the cDNA sequences with their corresponding genomic sequences. The Pfam database was used to investigate the PRMT conserved domains with default parameters. The conserved motifs of maize PRMT proteins encoded by putative *ZmPRMT* genes were analyzed by using MEME (https://meme-suite.org/meme/tools/meme) (accessed on 12 September 2021) online website with full-length protein sequences [67].

### 4.4. Chromosome Localization and Prediction of Cis-Acting Elements in Maize PRMT Genes

To investigate the position of genes on chromosomes and their possible functions, we completed the localization and cis-acting element analysis. The chromosome location of maize *PRMT* genes was obtained from the maize genomic database. The online mapping tool MG2C (http://mg2c.iask.in/mg2c_v2.0/) (accessed on 13 September 2021) is used to generate chromosome localization image of maize *PRMT* genes. In addition, the cis-acting elements in the promoter sequences can be involved in regulating gene transcription in abiotic stress responses, which depends on binding to different transcription factors. The 2 kb sequences upstream of the initiation codon (ATG) of each *ZmPRMT* gene were downloaded from the phytozome database (https://phytozome.jgi.doe.gov/pz/portal.html) (accessed on 13 September 2021) to further investigate the potential regulatory elements. The stress- and hormone-related cis-acting regulatory elements in promoter regions of *PRMT* genes in maize were analyzed by using the PlantCARE database (http://bioinformatics.psb.ugent.be/webtools/plantcare/html) (accessed on 13 September 2021) [68].

### 4.5. Expression Profile Analysis of Maize PRMT Genes in Developmental Tissues and Response to Abiotic Stress

To investigate the spatiotemporal expression patterns of *ZmPRMT* genes, the previously reported whole genome gene expression data of maize inbred line B73 was used to analyze the tissue-specific expression patterns of *ZmPRMT* genes as shown in Appendix A [52]. Using Treeview software (http://jtreeview.sourceforge.net/) (accessed on 15 September 2021) and Cluster 3.0 (http://bonsai.hgc.jp/mdehoon/software/cluster/software.htm) (accessed on 15 September 2021), we performed the hierarchical clustering analysis to investigate the corresponding gene expression patterns at different developmental stages of maize, according to the Pearson coefficients with average linkage. The result was further visualized to obtain the heat map. To further confirm the influence of external environmental factors on the expression of *ZmPRMT* genes in maize, the qRT-PCR expression profiles of leaves, stems and roots after abiotic stress treatment were analyzed. The several four-week B73 seedlings under similar growth status were transferred to the incubator of 42 °C and sampled the roots, stem and leaf of 0 h, 1 h, 2 h, 4 h and 8 h (0 h as the control groups). To ensure the reliability of the quantitative analysis, at least 3 biological repeats were completed. Total RNA was extracted from the collected tissues via Trizol reagent (Invitrogen, USA) and was purified to remove genomic DNA contamination using DNase I. Reverse transcription reactions were performed using 1μg of total RNA by the QuantiTectRev. Transcription Kit (Qiangen, Germany). The qRT-PCR reverse transcription reactions were carried out using a CFX96™ Real-Time PCR Detection System (Bio-Rad Laboratories, Inc., USA). Each cDNA sample was biologically replicated at least three times. The maize *Actin**1* gene was used as a standardized internal control. The relative mRNA levels of PRMT genes were calculated by using 2^−ΔΔCT^ method to compare the fold changes of related gene expression levels. The total primers in this study were listed in Appendix A.

### 4.6. Vector Construction and Arabidopsis Genetic Transformation

To verify the biological roles of *ZmPRMT* genes, the transgenic *Arabidopsis* plants overexpressing *ZmPRMT1* gene was constructed as a typical representative to explore functional roles in plants. The plasmid was constructed by amplifying the *ZmPRMT1* gene using PCR primers carrying proper restriction enzyme sites. The amplified *ZmPRMT1* cDNA was cloned into PHB vector driven by 35S promoter. The primers with enzyme sites are shown in Appendix A. The vectors were transformed into the Agrobacterium strain for transformation into *Arabidopsis* by the floral dip method. Seeds of the T0 and T1 generations were selected on 1/2 MS plates supplemented with hygromycin (30 ug/mL) and confirmed by PCR with primers (Hyg-F: GGTCGCGGAGGCTATGGATGC; Hyg-R: GCTTCTGCGGGCGATTTGTGT). The presence of *ZmPRMT1* was verified by qRT-PCR using specific primers. Three independent homozygous overexpression lines 2, 6 and 7 were selected, and the expression level of *ZmPRMT1* gene was quantified. At least 3 biological repeats were taken for all samples to ensure the reliability of the results of qRT-PCR analysis.

### 4.7. Flowering Time Assessment of Transgenic Arabidopsis

To verify the effect of *ZmPRMT1* gene on flowering time of *Arabidopsis*, we observed the phenotypic changes of *Arabidopsis* before and after flowering. Seedlings of WT and transgenic *Arabidopsis* were grown in the temperature incubator at 22 °C under day-long conditions (16 h light and 8 h dark) and 60% relative humidity. Flowering time of *Arabidopsis* was calculated by counting the number of rosette leaves after flowering. At least 3 plants were counted for each line.

### 4.8. Heat Tolerance Assay of Transgenic Arabidopsis

To verify the function of *ZmPRMT1* gene on heat tolerance of *Arabidopsis*, we conducted heat stress treatment on transgenic *Arabidopsis.* The seeds of *Arabidopsis* were sterilized with 75% alcohol and then vernalized at low temperature on the plate. Then the sprouted Arabidopsis seedlings are transplanted into the soil. Seedlings of *Arabidopsis* were also grown in the temperature incubator at 22 °C with a growth photoperiod of 16 h light and 8 h dark and 60% relative humidity. After 21 days, seedlings were exposed to heat stress (42 °C) for 8 h. Seedlings grown in the dark incubator at 22 °C with enough water were used as the control, and seedling leaves were collected after heat stress treatment and stored at −80 °C with liquid nitrogen to isolate RNA.

### 4.9. RNA Extraction and Quantitative Real-Time PCR Analysis of Transgenic Arabidopsis

To verify the biological function of *ZmPRMT1* gene in the transgenic *Arabidopsis*, we conducted qRT-PCR experiments to verify the expression profile of related genes. Based on the manufacturer’s instruction, total RNA was isolated from the collected samples using Trizol RNA isolation (USA). For quantitative real-time PCR, cDNA from three distinct biological samples was used for analysis. The PCR condition was carried out as follows: pre-denaturation for 15 min at 95 °C, 40 cycles at 95 °C for 10 s, 55 °C for 30 s and 72 °C for 30 s. The gene expression levels were calculated using the 2^−^^△△Ct^ method [69]. Each experiment was conducted in the form of at least three technologies and biological replication. The total primers in this study were listed in Appendix A.

### 4.10. Physiological Parameter Determination

In order to explore the function of *ZmPRMT1* gene on the change of proline accumulation under heat stress, we conducted physiological experiments to determine the change of proline accumulation. The content of proline was measured using the proline assay kit (Nanjing Jiancheng, Nanjing, China). The principle of determination is that in plants, only proline can react with acid ninhydrin to produce stable red compounds. The product has a maximum absorption peak at 520 nm, and its absorption value is linearly related to the content of proline. The testing tube, contrast tube and blank tube were boiled about 30 min, and the absorbance was measured on a spectrophotometer at 520 nm using double-distilled water as a standard. The experiment was performed for repeating at least three times.

## 5. Conclusions

In conclusion, we identified eight *ZmPRMT* genes encoding protein arginine methyltransferases in this study. The classification, evolutionary relationship, conserved motifs and domains, gene structure and stress-responsive cis-regulatory elements in promoter regions of *ZmPRMT* genes were determined to provide insights into their potential functions in this study. The comprehensive expression profiles of all maize *PRMT* genes in different tissues revealed that these genes may play functional roles at diverse developmental stages of maize. Furthermore, the expression levels of *ZmPRMT* genes were upregulated or downregulated under three various abiotic stress treatments, implying their potential roles in abiotic stress responses. What’s more, the construction of transgenic *Arabidopsis* lines further demonstrated the *ZmPRMT1* gene may play a crucial role in flowering time regulation and heat resistance. Compared with the WT plants, the transgenic *Arabidopsis* plants exhibited early flowering by modulating the expression of flowering-related regulatory genes, which might result in alteration in expression levels of floral meristem identity genes. In addition, the transgenic *Arabidopsis* exhibited enhanced heat tolerance through modulating proline metabolism related genes, which might result in changes in proline accumulation. Finally, via analyzing the expression levels of *ZmPRMT1* gene in flowering process and under heat stress treatment, we conclude that *ZmPRMT1* may play a critical role in growth and development processes, as well as abiotic stress responses. Taken together, the exploration of the maize *PRMT* gene family in organization, structure, evolution, expression profiling and the preliminary functional verification of *ZmPRMT1* will facilitate future functional verification of *ZmPRMT* genes and provide an important theoretical basis for a better comprehension of molecular epigenetic mechanism of regulating flowering and adaptation to abiotic stresses in maize. What’s more important, this study provides new insights and ideas for applying epigenetic methods to crop breeding.

## Figures and Tables

**Figure 1 ijms-23-12793-f001:**
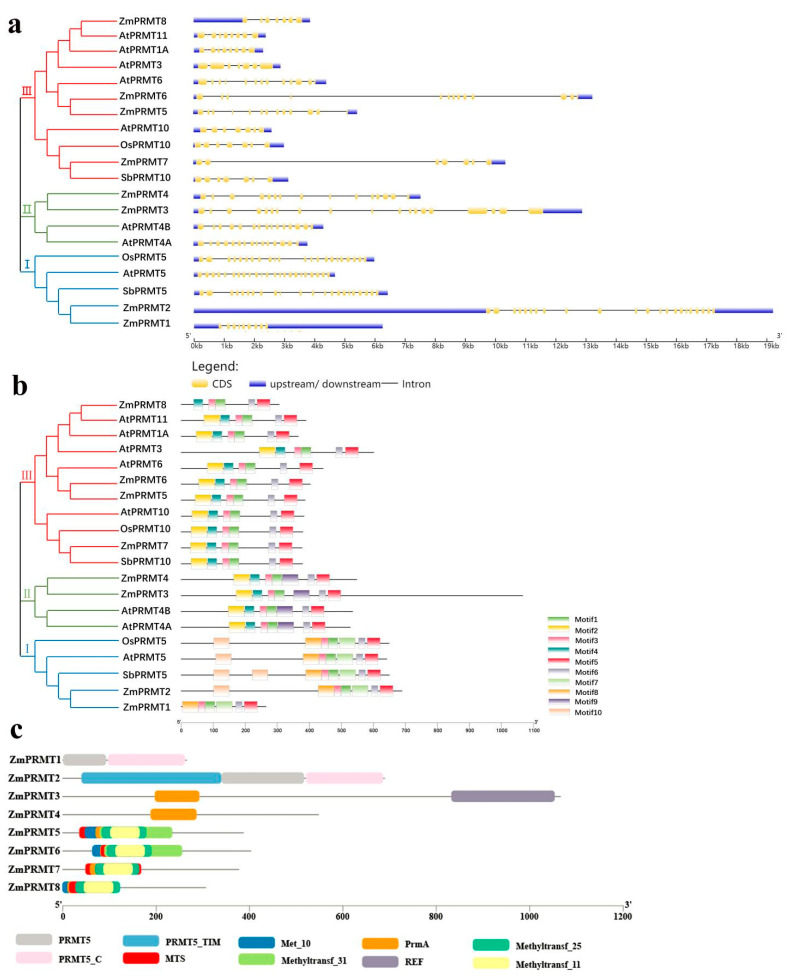
Phylogenetic relationship, gene structure, conserved motif and conserved domain analysis of maize PRMT family from maize, rice, sorghum and *Arabidopsis*. (**a**) Phylogenetic tree and the corresponding exon–intron structures of PRMT proteins. The maximum-likelihood phylogenetic tree was constructed using MEGA 7.0 with 1000 replicates. Three main clades are marked: I, II and III with different colored ranges. In the exon–intron structures, black lines represent introns, yellow boxes represent exons and the blue boxes represent upstream/downstream regions of PRMT genes. Protein sequences were downloaded from the Maize genome database and NCBI database. (**b**) Phylogenetic tree and the corresponding conserved motifs of PRMT proteins. Ten different colored boxes with numbers are used to represent different conserved motifs. The motifs identified in each group of PRMT proteins were schematically represented using the MEME motif search tool. (**c**) Phylogenetic tree and the corresponding conserved domains of maize PRMT proteins. The 8 PRMT domain sequences of maize were downloaded from the Smart database. Protein is represented by gray line. The position to the left of the gray line represents the N-terminal of each protein. Different colored rectangles are used to represent the domains contained in proteins.

**Figure 2 ijms-23-12793-f002:**
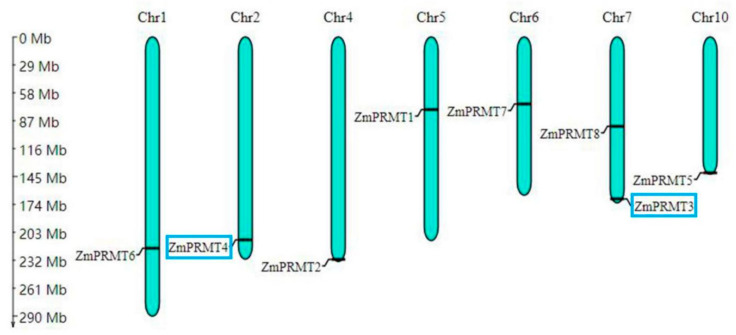
Chromosomal location of PRMT family genes in maize. Eight maize *PRMT* genes were located on 7 of all the 10 maize chromosomes. The number of chromosomes is marked at the top of each blue column bar. The approximate position of each *ZmPRMT* gene on the chromosome corresponds to the gene name on the left of the blue bar in the figure. The scale on the left in the figure is in megabytes. The blue box indicates that there is a replication relationship between genes.

**Figure 3 ijms-23-12793-f003:**
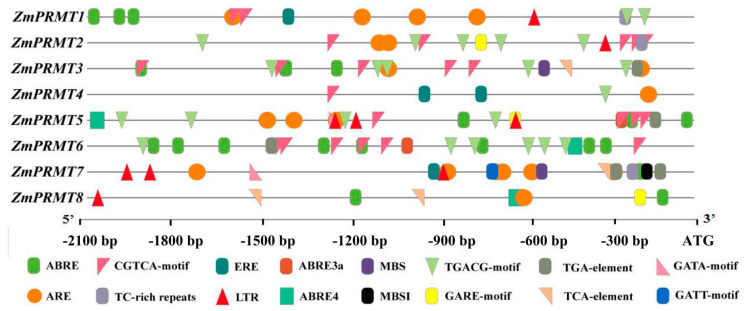
Distribution of major stress-related cis-elements in the promoter sequences of the 8 *ZmPRMT* genes. Putative ABRE, CGTCA-motif, ERE, MBS, TGACG-motif, GATA-motif, TGA-element, ARE, TC-rich repeats, LTR, GARE-motif, TCA-element and GATT-motif core sequences are represented by different symbols as shown in the symbols at the bottom. ABRE: cis-acting element involved in the abscisic acid responsiveness; CGTCA-motif: cis-acting regulatory element involved in the MeJA responsiveness; ERE: ethylene-responsive element; MBS: drought -responsive element; TGACG-motif: cis-acting regulatory element involved in the MeJA-responsiveness; GATA-motif: zinc-finger transcription factor; TGA element: auxin-responsive element; ARE: anaerobic-responsive element; TC-rich repeats: cis-acting element involved in defense and stress responsiveness; LTR: low-temperature relative element; GARE-motif: gibberellin-responsive element; TCA-element: cis-acting element involved in salicylic acid responsiveness. The position of each cis-element in the 2 kb sequence upstream of the initiation codon (ATG) of the *ZmPRMT* genes was measured at a scale of 300 bp.

**Figure 4 ijms-23-12793-f004:**
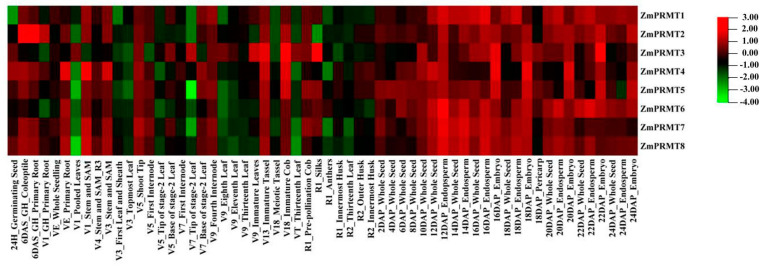
Hierarchical cluster analysis of expression profiles of 8 maize *ZmPRMT* genes. Hierarchical cluster analysis of expression profiles of *ZmPRMT* genes family in all 60 tissues that including the whole growth and development process of maize. The color code on the right indicates the log2 signal value. The gene name of each *ZmPRMT* is shown on the left of each line. Red represents a high level and green indicates a low level of transcript abundance. The tissues and/or organs in different development periods are noted on the bottom of each lane.

**Figure 5 ijms-23-12793-f005:**
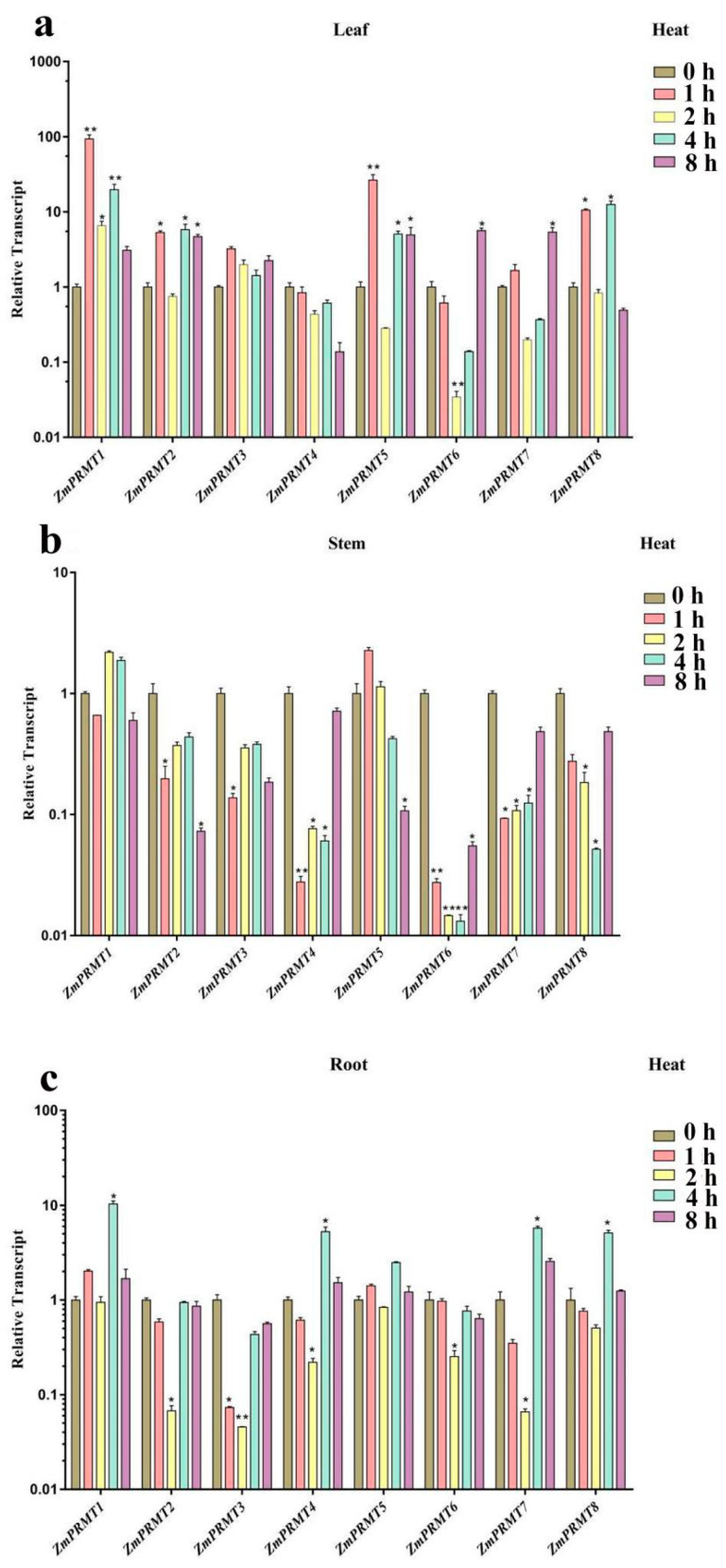
Expression profiling of the 8 *ZmPRMT* genes exposed to heat stress treatment in maize tissues (roots, stems and leaves). (**a**) The expression profiling of the 8 *ZmPRMT* genes exposed to heat stress treatment in maize leaves. (**b**) The expression profiling of the 8 *ZmPRMT* genes exposed to heat stress treatment in maize stems. (**c**) The expression profiling of the 8 *ZmPRMT* genes exposed to heat stress treatment in maize roots. The qRT-PCR data was standardized with the maize *ZmActin1* gene. The control condition was represented as CK (0 h), and the heat stress conditions were represented as Heat (1 h, 2 h, 4 h and 8 h). X-axes represent the genes under different treatments, while y-axes represent the scale of the relative expression level of genes. Error bar is generated by three biological repetitions. The significance level is indicated by the asterisk at the top of the error bar. *p* < 0.05 is significantly different *; *p* < 0.01 is significantly different **.

**Figure 6 ijms-23-12793-f006:**
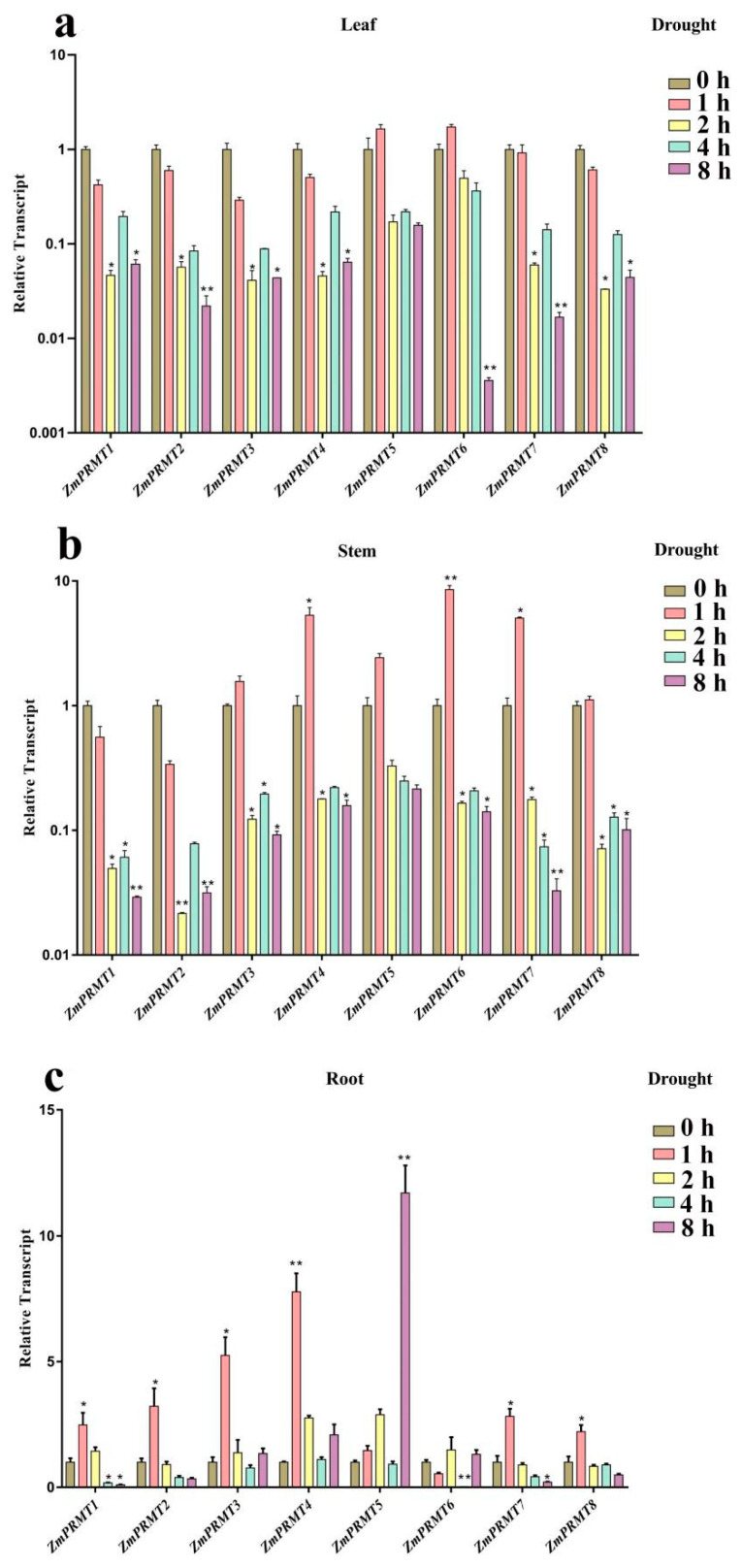
Expression profiling of the 8 *ZmPRMT* genes exposed to drought stress treatment in maize tissues (roots, stems and leaves). (**a**) The expression profiling of the 8 *ZmPRMT* genes exposed to drought stress treatment in maize leaves. (**b**) The expression profiling of the 8 *ZmPRMT* genes exposed to drought stress treatment in maize stems. (**c**) The expression profiling of the 8 *ZmPRMT* genes exposed to drought stress treatment in maize roots. The qRT-PCR data was standardized with the maize *ZmActin1* gene. The control condition was represented as CK (0 h), and the heat stress conditions were represented as Drought (1 h, 2 h, 4 h and 8 h). X-axes represent the genes under different treatments, while the y-axes represent the scale of the relative expression level of genes. Error bar is generated by three biological repetitions. The significance level is indicated by the asterisk at the top of the error bar. *p* < 0.05 is significantly different *; *p* < 0.01 is significantly different **.

**Figure 7 ijms-23-12793-f007:**
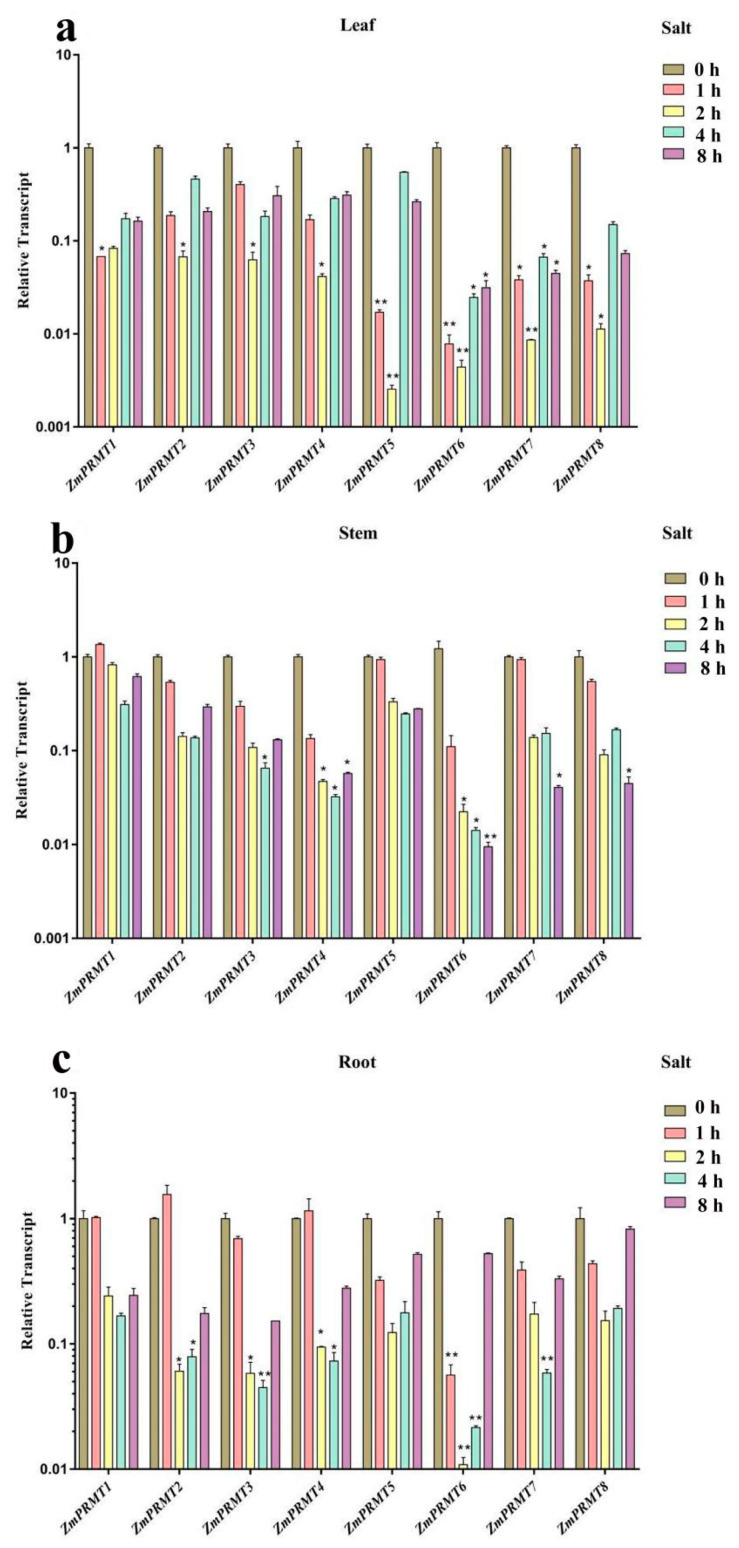
Expression profiling of the 8 *ZmPRMT* genes exposed to salt stress treatment in maize tissues (roots, stems and leaves). (**a**) The expression profiling of the 8 *ZmPRMT* genes exposed to salt stress treatment in maize leaves. (**b**) The expression profiling of the 8 *ZmPRMT* genes exposed to salt stress treatment in maize stems. (**c**) The expression profiling of the 8 *ZmPRMT* genes exposed to salt stress treatment in maize roots. The qRT-PCR data was standardized with maize *ZmActin1* gene. The control condition was represented as CK (0 h), and the heat stress conditions were represented as Salt (1 h, 2 h, 4 h and 8 h). X-axes represent the genes under different treatments, while y-axes represent the scale of the relative expression level of genes. Error bar is generated by three biological repetitions. The significance level is indicated by the asterisk at the top of the error bar. *p* < 0.05 is significantly different *; *p* < 0.01 is significantly different **.

**Figure 8 ijms-23-12793-f008:**
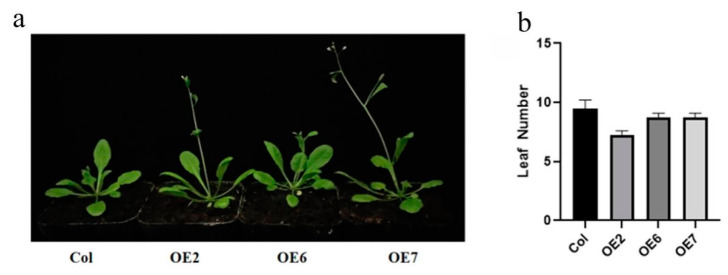
Regulation of flowering time and leaf number by *ZmPRMT1* in transgenic *Arabidopsis* lines. (**a**) The early-flowering phenotype of transgenic *Arabidopsis* lines was observed. The left figure shows the phenotype of the WT and transgenic *Arabidopsis* lines 2, 6 and 7. Seedlings of these 4 different *Arabidopsis* lines were grown in the temperature incubator at 22 °C under long-day conditions (16 h light and 8 h dark) and 60% relative humidity. After 27 days, 4 different *Arabidopsis* lines flowering successively. (**b**) Total leaf number of transgenic *Arabidopsis* lines was calculated. The right figure shows leaf number of the corresponding plants left. Total leaf number of WT, transgenic *Arabidopsis* 2, 6 and 7 was calculated under same conditions.

**Figure 9 ijms-23-12793-f009:**
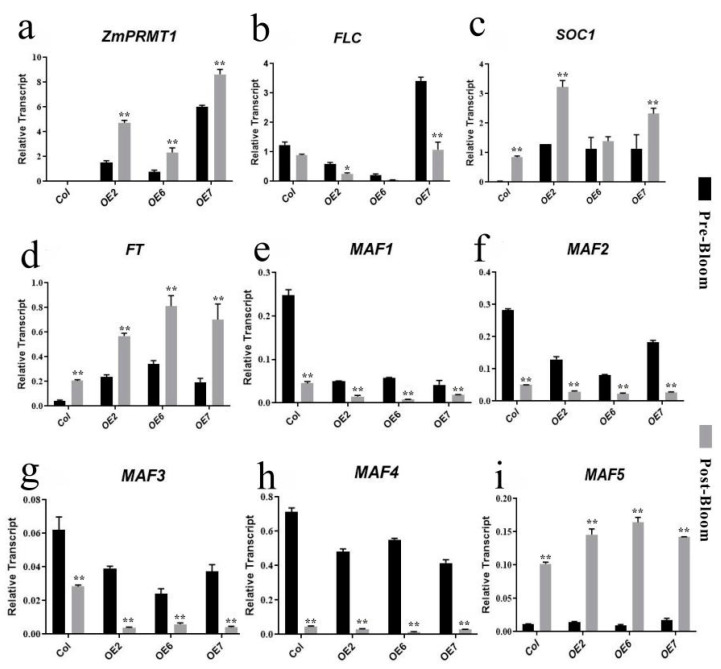
Expression of flowering-related genes by qRT-PCR analysis. The flowering-related genes, including (**b**) FLC (AT5G10140), MADS BOX PROTEIN FLOWERING LOCUS F; (**c**) SOC1 (AT2G45660), SUPPRESSOR OF OVEREXPRESSION OF CONSTANS 1; (**d**) FT (AT1G65480), FLOWERING LOCUS T; (**e**) MAF1 (AT1G77080), MADS AFFECTING FLOWERING 1; (**f**) MAF2 (AT5G65050) MADS AFFECTING FLOWERING 2; (**g**) MAF3 (AT5G65060) MADS AFFECTING FLOWERING 3; (**h**) MAF4 (AT5G65070) MADS AFFECTING FLOWERING 4; (**i**) MAF5 (AT5G65080) MADS AFFECTING FLOWERING 5. The mRNA levels were determined using qRT-PCR, and the values are reported relative to *ZmPRMT1* mRNA levels in each line (**a**). The expression levels of related genes before flowering are represented by black columns, and the expression levels of related genes after flowering is represented by gray columns. Error bar is generated by three biological repetitions. The significance level is indicated by the asterisk at the top of the error bar. *p* < 0.05 is significantly different *; *p* < 0.01 is significantly different **.

**Figure 10 ijms-23-12793-f010:**
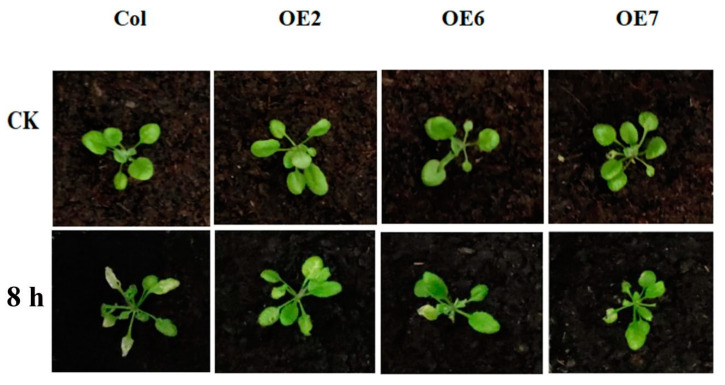
Heat tolerance analysis of transgenic *Arabidopsis* lines. The phenotypes of the WT and transgenic *Arabidopsis* seedling lines 2, 6 and 7 are shown following their treatment with different heat stress. CK (0 h), the control condition and Heat (8 h), the heat stress conditions. The 3-week-old seedlings were directly exposed to 42 °C to detect heat tolerance in 0 h and 8 h. The *Arabidopsis* were photographed in 3 d after the 0 h and 8 h heat stress treatment. The wilting degree of leaves of both the WT and transgenic lines was observed at 3 days after the 8 h heat stress treatment.

**Figure 11 ijms-23-12793-f011:**
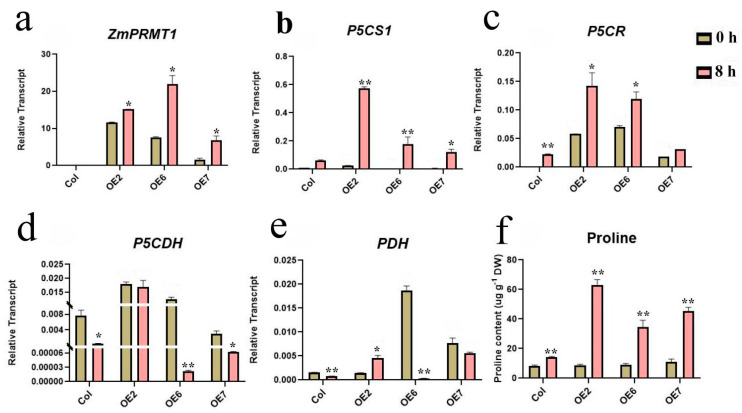
Analysis of heat tolerance-related genes expression and proline accumulation under heat stress by qRT-PCR. The plant abiotic stress tolerance-related genes, including genes in the proline synthesis way such as (**b**) P5CS1 (AT2G39800), delta1-pyrroline-5-carboxylate synthase 1; (**c**) P5CR (AT5G14800), pyrroline-5-carboxylate (P5C) reductase and genes in the proline degradation way such as (**d**) P5CDH (AT5G62530), DELTA1-PYROLINE-5-CARBOXYLATE DEHYDROGENASE; (**e**) PDH (AT3G30775) and PROLINE DEHYDROGENASE. The mRNA levels were determined using qRT-PCR, and the values are reported relative to *ZmPRMT1* mRNA levels in each line (**a**). The proline accumulation contents were shown in (**f**). Error bar is generated by three biological repetitions. The significance level is indicated by the asterisk at the top of the error bar. *p* < 0.05 is significantly different *; *p* < 0.01 is significantly different **.

**Figure 12 ijms-23-12793-f012:**
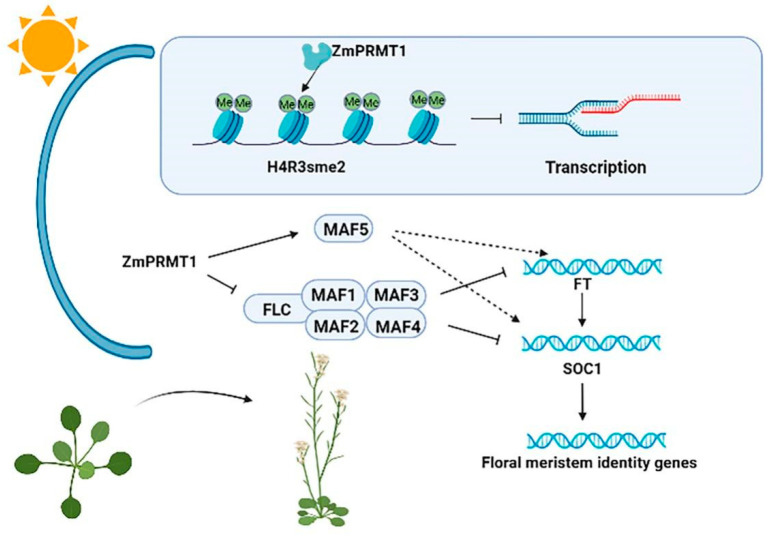
Hypothetical model for the regulation of flowering time controlled by *ZmPRMT1* in *Arabidopsis*. Overexpression of *ZmPRMT1* in *Arabidopsis* leads to downregulation of the expression levels of *FLC* and *MAF* genes (*MAF1-MAF4*), and its products assembled into MADS-box repressor complexes. However, the expression levels of *MAF5* were upregulated with an unclear manner in plant growth process. FLC and MADS domain complex forms large protein complexes at the *FT* and *SOC1* sites to repress the expression of these two genes. H4R3sme2 is one of the possible targets for *ZmPRMT1* to function. The arrows indicate positive modulation, and the “T” bars indicate negative modulation. The dashed lines represent assumed interactions.

**Figure 13 ijms-23-12793-f013:**
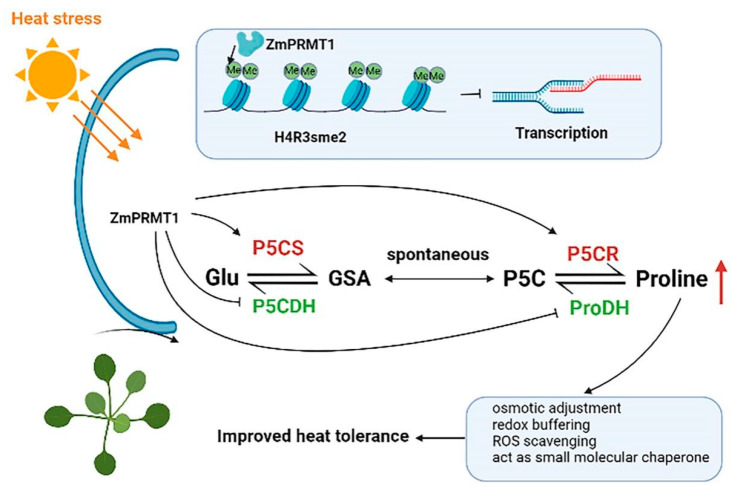
Hypothetical model showing the overexpression of *ZmPRMT1* results in improved resistance under heat stress in *Arabidopsis* involving proline metabolism pathway. The pathway of proline metabolism. Glu glutamic acid, GSA glutamate-1-semi-aldehyde, P5C pyrroline-5-carboxylate, P5CS1 delta-1-pyrroline-5-carboxylate synthase1, P5CR pyrroline-5-carboxylate reductase, P5CDH delta-1-pyrroline-5-carboxylate dehydrogenase, ProDH proline dehydrogenase. Overexpression of *ZmPRMT1* in *Arabidopsis* leads to upregulation of the expression levels of *P5CS1* and *P5CR* gene in proline synthesis pathway while downregulation of the expression levels of *P5CDH* and *PDH* gene in the proline degradation pathway. Finally, proline accumulation increased. H4R3sme2 is one of the possible targets for *ZmPRMT1* to function. The upregulation of gene expression levels is represented by red font, whereas downregulation of gene expression levels is represented by green font. The arrows indicate positive modulation, and the “T” bars indicate negative modulation. The dashed lines represent assumed interactions.

**Figure 14 ijms-23-12793-f014:**
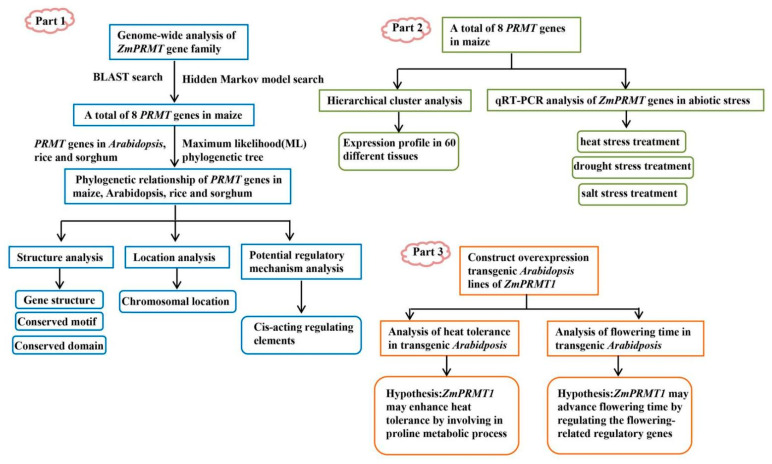
The schematic flowchart of the study.

**Table 1 ijms-23-12793-t001:** Basic information about ZmPRMT in maize.

Gene Name	Accession Number Ensemble Transcript	Genome Location Coordinates (5′-3′)	CDS (bp)	Protein			
				Length (a.a)	Mol. Wt (kDa)	PI	Chr
							No.
ZmPRMT1	Zm00001d015228	80075042–80104640	798	265	29.86	5.761	5
ZmPRMT2	Zm00001d054001	244881916–244901087	2070	689	77.33	5.618	4
ZmPRMT3	Zm00001d022469	178246469–178259200	3201	1066	115.42	7.413	7
ZmPRMT4	Zm00001d007133	223062728–223070243	1647	548	60.44	5.127	2
ZmPRMT5	Zm00001d026614	148995156–149000603	1164	387	43.65	5.274	10
ZmPRMT6	Zm00001d032633	232602048–232615275	1212	403	44.97	5.971	1
ZmPRMT7	Zm00001d036131	73847683–73858041	1134	377	42.37	5.423	6
ZmPRMT8	Zm00001d020188	98545533–98549295	921	306	34.52	6.862	7

## Data Availability

Not applicable.

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
