# Peer review of "Genome-Wide Identification of Maize Protein Arginine Methyltransferase Genes and Functional Analysis of ZmPRMT1 Reveal Essential Roles in Arabidopsis Flowering Regulation and Abiotic Stress Tolerance"

_ijms, 2022, doi:10.3390/ijms232112793_

Round 1

Reviewer 1 Report

The article entitled " Genome-wide identification and functional analysis of protein arginine methyltransferase genes in maize reveal essential roles in flowering regulation and abiotic stress tolerance". The authors have performed the identification and functional analysis of protein arginine methyltransferase (PMRT) genes in maize with series of bioinformatic analyses. qRT-PCR analysis revealed these genes are responsible for heat, drought and salinity stress treatments. Further, authors have over expressed the ZmPRMT1 gene to Arabidopsis and the results suggested that it can promote the earlier flowering time through altering the proline accumulation in transgenic Arabidopsis plants. On the whole, the study dissected the role of ZmPMRT gene in the regulation of flowering and abiotic stresses in maize.

The manuscript comprises all the necessary elements of scientific novelty. The experimental designing and execution of the study were appreciable. I recommend this article for publication after incorporating changes given in below.

Abstract need to be concise with findings.

Line 19-20: Reframe it.

Introduction is comprehensive and according to the studies issue. But it can be concise little.

Line 74: write scientific name of the plants which you have mentioned.

Line 114,116: Reference should be numbered.

Authors must concentrate on the formatting, and use of symbols, etc., All the gene names should be in italics. check the same throughout the manuscript.

In materials and method section: authors should explain why each item of methodology was done.

Framework figure is required. It will be useful to the readers for better understanding of the studied issue.

In results section, the image resolutions are not good. Please enhance the resolution.

Line 360: It should be qRT-PCR.

The p value ‘p’ should be in italics.

If possible, merge the sections 2.6-2.8.

What about the gene (ZmPRMT1) linked recognition site of the Hind3/Xba1 primer.

Please remove the subheadings in the discussion section.

If possible, authors are advised to perform comparative analysis with ZmPMRT to Oryza sativa and Sorghum bicolor PMRT genes. Use Gramene database for identification and visualize it by Circos tool. It will provide added value to your MS.

Conclusion section needs to be improvised and add few lines about future perspectives and hypothesize the current study. It will be useful to the readers community to design and understand the importance of studied issue.

Reviewer 2 Report

Dear authors,

Manuscript ijms-1966559 entiteled "Genome-wide identification and functional analysis of protein arginine methyltransferase genes in maize reveal essential roles in flowering regulation and abiotic stress tolerance" and authored by Qiqi Ling , Jiayao Liao , Xiang Liu , Yue Zhou and Yexiong Qian targets a hot topic and is of huge interest to the journal readers. Unfortunately while the research design have is accurate and the experiments nicely conducted several points needs to be addressed before the paper can be recommended for publication:

1. I do not agree about the affirmation of the title : the essential roles in flowering regulation and abiotic stress tolerance is in Arabidopsis and not in maize ! you have to be very careful because this is speculation ! you don't have any evidence at this stage that the genes have the same function in maize otherwise please provide data about overexpression of the gene in maize plants! so please change the title to reflect the results of the study without speculation

2. In the introduction section : please discuss how expression of maize genes in Arabidopsis could reflect the function of the gene. To which level we can rely on expression of maize genes in Arabidopsis to discover gene functions in maize and highlight the limits of this strategy ! litterature is very rich with such examples ! This is a critical issue for your conclusions so please deeply argument this part !

3. The quality of presentation of figures is very weak ! the figures are actually non readable to me ! Please meet the journal standards regarding this point ! without addressing this point I could not recommend your manuscript for publication.

Finally I am waiting to read an improved version of this manuscript that I could recomnmend for publication.

Best regards

Round 2

Reviewer 2 Report

Dear authors,

I believe your manuscript is now ready for acceptance

Best regards